# OpenReview forum: "Refuse Whenever You Feel Unsafe: Improving Safety in LLMs via Decoupled Refusal Training"
_ICLR.cc/2025/Conference — ICLR 2025 Conference Withdrawn Submission_

### Official Review · Reviewer_CJf6 · 2024-10-15

**Soundness:** 2
**Presentation:** 3
**Contribution:** 3
**Rating:** 5
**Confidence:** 3

**Summary:**

This paper proposes a new training strategy to help the model better refuse harmful requests. Specifically, it introduces two novel components: (1) MLE with harmful response prefix, and (2) Reinforced Transition Optimization. By integrating these two parts into the fine-tuning objectives, the authors observe significant ASR reduction across several state of the art safety benchmarks, while the model's utility maintains the same as the vanilla one.

**Strengths:**

1. The author noticed the issue that the current standard safety training strategy has a refusal position bias, which may lead the model cannot correctly generate refusal once it starts to comply with the request at the first stage.
2. The author proposed a novel safety training strategy that can mitigate this bias and the reported results show some promise compared to the the vanilla training.

**Weaknesses:**

1. "Refuse at a later stage" is not a good indicator to show the model's safety. Ideally, for a well-aligned model, when it faces a harmful request, it **should** refuse at the first stage, instead of generating a refusal after the request has been **fully satisfied** (For example, if you ask the model how to make a bomb, it first generate step-by-step guidance, then generate refusal. Obviously, we cannot treat this as a successful defense). Though the case studies reported in the paper do not show this issue, I still doubt this may happen based on my understanding of the proposed training mechanisms. Authors could provide statistics on how often the model generates any harmful content before refusing, or provide ablation studies to show/prove that it won't happen.
2. The experiment results reported in the paper do not show how the randomness will affect the results. To be more specific, I have no idea whether these results come from a single experiment, or are gathered across several repetitions. The randomness of dataset shuffling during the training process, as well as the randomness brought from the sampling strategy for model generation, will probability introduce a significant randomness to the final results. The authors should also report these details, and report confidence intervals for all the results if necessary.

**Questions:**

1. I don't understand why we need to use 6000 examples for vanilla fine-tuning for fair comparison. Based on my understanding, the triple of (harmful query, safe response, harmful response) should be treated as one example instead of two, given that the final training objective for DeRTa incorporates both harmful response and safe response.
2. Mistral has several versions (v0.1, v0.2), which version is used for experiments?
3. For the training objective of DeRTa, $\hat{r}_{<k}$, where $k$ is randomly sampled from 0 to $|\hat{r}|$, which means the harmful prefix will be randomly truncated to a certain length and directly concatenated to the safety response. I doubt it may hurt the fluency of the final generation. I am wondering whether the authors have observed this phenomenon. If so, an easy fix is to truncate the harmful prefix only after the sentence is complete.
4. For safety evaluation, how is the ASR computed? Keyword? Human labeling? or GPT-judge?
5. In Appendix A, for the experiment setup, do the vanilla training and DeRTa training share the same set of hyperparameters?

---

### Official Review · Reviewer_S5C1 · 2024-10-30

**Soundness:** 1
**Presentation:** 2
**Contribution:** 1
**Rating:** 3
**Confidence:** 4

**Summary:**

This paper studies the safety tuning of LLMs to enhance their safety. The key idea is to append a piece of harmful responses as prefix, and use RTO to fine-tune the models such that the models can refuse the prompt at any position. The authors show that the proposed method can ensure safety while preserving utility in downstream tasks.

**Strengths:**

The paper investigates safety tuning of LLMs, and proposes a method to encourage LLMs to refuse prompts at any position of responses.

**Weaknesses:**

- The motivation of this paper is not clear and should be demonstrated using preliminary experimental results.
- The technical contribution of this paper is incremental. The method applies existing machine learning techniques as solutions to solve the problem. The authors could clarify/highlight the novelty and contribution further.
- No baseline comparison can be found in experiments, making it hard to judge the effectiveness of proposed method.

**Questions:**

- The motivation of the paper should be further clarified. I agree with motivation (2) in line 75, but it is hard to follow motivation (1). In Fig. 2(a), the response seems to be fine and safe. Shouldn't human values prefer the response in 2(a) rather than 2(b)? To strengthen the motivation, especially (2), the authors could perform some preliminary analysis to demonstrate that SOTA **instruct** models generate harmful responses under attacks, even when the first few tokens of responses decline the harmful instructions.
- It is unclear how exactly refusal position bias is defined. Table 1 can be expanded to incorporate more models as justification to the claims in line 145 - 150.
- There are a wide range of refusal tokens, but equation (2) only uses "sorry". Will this limit the performance of proposed method?
- The experimental results are evaluated manually by humans. The authors could also consider GPT judges to avoid biases from humans. Cross evaluating the responses by multiple GPT judges could also mitigate biases from GPTs.
- There is no baseline comparison in the experiments. The authors listed multiple existing defenses in related work, but none of them is compared in any experiment. For example, I doubt if the proposed method can outperform other defenses under PAIR.
- The proposed method should be evaluated on more model families, even for small models from families such as Phi and Qwen.
- The experiments only demonstrate the safety by comparing with Vanilla model. It is more meaningful to evaluate the safety performance on instruct models such as Llama3-8B-Instruct or Mistral-Instruct-v3.
- I'm curious of how the proposed method would perform under GCG attack, which is only evaluated on small models. The authors could discuss potential challenges or limitations when applying GCG to larger models such as Llama3-8B or Llama3-70B.
- Is there any explanation on why helpfulness could increase after fine-tuning with refusal responses?
- How should one understand the results in Fig. 4? It seems the proposed method has lower refusal ratio compared to vanilla models, or am I missing something in this figure?

---

### Official Review · Reviewer_yN4q · 2024-11-04

**Soundness:** 3
**Presentation:** 2
**Contribution:** 2
**Rating:** 5
**Confidence:** 4

**Summary:**

The paper presented a tuning method to enable a white-box large language model to defense jailbreak attack. Harmful Response Prefix was applied to help model recognize unsafe content at any response position, as Reinforced Transition Optimization would help generate safety refusal at every position.

**Strengths:**

1. The experiments seems helpful to defend different types of jailbreak attacks;

**Weaknesses:**

1. The paper is lack of the comparison with other SOTA methodology against jailbreak attacks; The paper will be more persuasive if more well-designed experiments are presented;

**Questions:**

The jailbreak attack have greater chance to succeed in long text and multi-step dialogue, I am curious about the performace of your methods in those scenarios;

---

### Official Review · Reviewer_dQeb · 2024-11-06

**Soundness:** 3
**Presentation:** 3
**Contribution:** 2
**Rating:** 5
**Confidence:** 4

**Summary:**

Since refusal training typically places the refusal at the start of the prompt, there is a risk that models can be attacked as long as this first refusal is bypassed. This paper designs two training objectives that deal with this: outputting the refusal after a harmful response prefix, and outputting the word sorry for every token of a harmful response. The paper finds that this signifcantly decreases the ASR of standard jailbreaking attack methods.

**Strengths:**

1. This paper features a systematic evaluation of the method across various datasets and models. I am especially happy with the ablations on the contribution of each training objective and the multi-objective measurements across harmlessness and helpfulness.
2. The proposed ideas are simple and easy to implement.

**Weaknesses:**

1. I am not convinced by the strength of the baselines.
    - I can not seem to find where Vanilla- is defined in the paper
    - Why do some of the baselines get less data points?
    - How is the DPO fine-tuning set up? Specifically, does it also get access to completions that start harmful and turn into refusals?
    - In the related work, it is mentioned that prior work such as "safety shouldn't be a few tokens deep" had implemented a method quite similar to MLE with harmful prefix. Does your method perform better than this, or is the primary contribution of this paper the RTO algorithm?
2. There are no adaptive attacks in the paper designed to break this refusal mechanism. This is important to ensure that the method is actually more secure and not a simple modification to an adversary's algorithm (for example, the attacker could target a word not covered by the RTO optimization, or optimize harmful strings that also start with sorry).
3. I am worried about the novelty of this paper. As mentioned in the related work, the first part of the method is an existing component. The second component feels qualitatively similar to the first, just at every possible prefix length.

**Questions:**

1. In Figure 4, it seems like your method has less refusals than only harmful prefix and only vanilla. However, your method is supposed to have better ASR. How can these both be true?
2. I suggest citing this paper to motivate the defense: https://arxiv.org/abs/2404.02151
3.

---

### Official Review · Reviewer_cVUT · 2024-11-07

**Soundness:** 2
**Presentation:** 2
**Contribution:** 2
**Rating:** 3
**Confidence:** 4

**Summary:**

The authors introduce an approach which they call Decoupled Refusal Training (DeRTa) to improve Large Language Models (LLMs) safety by addressing the issue of refusal position bias in safety tuning data. The method consists of two main components: (1) Maximum Likelihood Estimation (MLE) with Harmful Response Prefix and (2) Reinforced Transition Optimization (RTO). The authors evaluate their approach on LLaMA3 and Mistral model families across various attack scenarios.

**Strengths:**

1. The paper identifies an important issue in current safety tuning practices - the refusal position bias
2. The proposed DeRTa method is reasonably motivated to address the identified problem
3. The ablations are fairly comprehensive across different attacks, LLMs, and benchmarks.

**Weaknesses:**

1. In the experiment setup, the authors report using randomly sampled examples but only report the average results. It remains unclear how sensitive the results are to the sampling. Furthermore, the authors should report or clarify whether the reported results are for one or multiple seeds.
2. Limited comparison with other safety-enhancing techniques: The paper does not thoroughly compare DeRTa with other recent safety-enhancing methods beyond standard safety tuning such as R2D2 [1].
3. The paper's main contribution is the identification of the refusal position bias in safety tuning data. However, the proposed solution does not represent a significant advancement in methodology. For instance, the addition of a Harmful Response Prefix is essentially an adaptation of data augmentation.
4. The DeRTa approach might lead to over-refusal, where LLMs refuse to generate even safe content due to the strong emphasis on safety. The authors should evaluate their approach against baselines in over-refusal, for instance using benchmarks like OR-Bench [2].

[1] HarmBench: A Standardized Evaluation Framework for Automated Red Teaming and Robust Refusal. Mantas Mazeika, Long Phan, Xuwang Yin, Andy Zou, Zifan Wang, Norman Mu, Elham Sakhaee, Nathaniel Li, Steven Basart, Bo Li, David Forsyth, Dan Hendrycks

[2] OR-Bench: An Over-Refusal Benchmark for Large Language Models. Justin Cui, Wei-Lin Chiang, Ion Stoica, Cho-Jui Hsieh

**Questions:**

1. How sensitive are reported results to the random sampling of examples from the evaluated benchmarks?
2. How does DeRTa compare to state-of-the-art safety-enhancing techniques on the evaluated benchmarks?
3. What is the over-refusal rate of DeRTa compared to the baselines?

---

### Note · Authors · 2024-11-21

I have read and agree with the venue's withdrawal policy on behalf of myself and my co-authors.